# Mycobacteria Tolerate Carbon Monoxide by Remodeling Their Respiratory Chain

Katherine Bayly,[a,b] Paul R. F. Cordero,[a,b] Ashleigh Kropp,[a] Cheng Huang,[c,d] Ralf B. Schittenhelm,[c,d] ![ORCID]Rhys Grinter,[a,b] ![ORCID]Chris Greening[a,b]

[a]Department of Microbiology, Monash Biomedicine Discovery Institute, Monash University, Clayton, VIC, Australia
[b]School of Biological Sciences, Monash University, Clayton, VIC, Australia
[c]Monash Proteomics and Metabolomics Facility, Monash Biomedicine Discovery Institute, Monash University, Clayton, VIC, Australia
[d]Department of Biochemistry, Monash Biomedicine Discovery Institute, Monash University, Clayton, VIC, Australia

Katherine Bayly and Paul R. F. Cordero contributed equally to this work. Author order was determined alphabetically.

**ABSTRACT** Carbon monoxide (CO) gas is infamous for its acute toxicity. This toxicity predominantly stems from its tendency to form carbonyl complexes with transition metals, thus inhibiting the heme-prosthetic groups of proteins, including respiratory terminal oxidases. While CO has been proposed as an antibacterial agent, the evidence supporting its toxicity toward bacteria is equivocal, and its cellular targets remain poorly defined. In this work, we investigate the physiological response of mycobacteria to CO. We show that *Mycobacterium smegmatis* is highly resistant to the toxic effects of CO, exhibiting only minor inhibition of growth when cultured in its presence. We profiled the proteome of *M. smegmatis* during growth in CO, identifying strong induction of cytochrome *bd* oxidase and members of the *dos* regulon, but relatively few other changes. We show that the activity of cytochrome *bd* oxidase is resistant to CO, whereas cytochrome *bcc-aa*$_3$ oxidase is strongly inhibited by this gas. Consistent with these findings, growth analysis shows that *M. smegmatis* lacking cytochrome *bd* oxidase displays a significant growth defect in the presence of CO, while induction of the *dos* regulon appears to be unimportant for adaptation to CO. Altogether, our findings indicate that *M. smegmatis* has considerable resistance to CO and benefits from respiratory flexibility to withstand its inhibitory effects.

**IMPORTANCE** Carbon monoxide has an infamous reputation as a toxic gas, and it has been suggested that it has potential as an antibacterial agent. Despite this, how bacteria resist its toxic effects is not well understood. In this study, we investigated how CO influences growth, proteome, and aerobic respiration of wild-type and mutant strains of *Mycobacterium smegmatis*. We show that this bacterium produces the CO-resistant cytochrome *bd* oxidase to tolerate poisoning of its CO-sensitive complex IV homolog. Further, we show that aside from this remodeling of its respiratory chain, *M. smegmatis* makes few other functional changes to its proteome, suggesting it has a high level of inherent resistance to CO.

**KEYWORDS** carbon monoxide, *Mycobacterium*, proteomics, respiratory oxidases

Carbon monoxide (CO) is notorious as a noxious gas, largely due to the acute toxicity observed in humans and higher vertebrates upon inhalation (1). However, while CO is an energy-rich molecule that can be oxidized to yield low-potential electrons, it is largely chemically inert under physiological conditions (2). The toxicity of CO stems from its tendency to form carbonyl complexes with transition metals (3), specifically the Fe ion of heme groups, which are indispensable for many biochemical processes (4). Despite considerable knowledge of the chemistry of CO-metal interactions, the

Address correspondence to Rhys Grinter, rhys.grinter@monash.edu, or Chris Greening, chris.greening@monash.edu.

specific cellular targets of CO toxicity are still only partially defined (5). Toxicity at a cellular level is thought to arise primarily from competitive inhibition of heme-containing respiratory enzymes, i.e., heme-copper terminal oxidases, though other targets are also probable given the abundance of heme-containing proteins in most cells (6).

While CO is acutely toxic to mammals, the evidence demonstrating this gas is also potently toxic toward bacteria is equivocal. Studies on diverse bacterial species have demonstrated that, while gaseous CO is inhibitory to bacteria, this inhibition is only observed at high CO partial pressures (2 to 30%), is transient, and acts to slow rather than halt bacterial growth (7–10). Studies treating bacteria with CO-releasing molecules (CORMs) have reported more acute toxicity and a bactericidal mode of action, which was attributed to CO (10, 11). However, subsequent analysis of the effect of CORMs strongly suggests that this toxicity is largely due to the transition metal complex that constitutes many CORMs, rather than toxicity of CO (12, 13). Further investigation is required to determine the extent to which CO is toxic to bacteria, the molecular targets of toxicity, and how bacteria can grow at high partial pressures of CO.

Our understanding of the role of CO in the physiology of *Mycobacterium* also remains incomplete. This diverse actinobacterial genus spans saprophytic species such as *Mycobacterium smegmatis* (14), as well as numerous human and animal pathogens, including *Mycobacterium tuberculosis* (15). Mycobacterial species, including *M. smegmatis* and *M. tuberculosis*, seem to be relatively resistant to CO and exhibit robust growth in the presence of a 20 to 30% CO atmosphere (8, 16). Furthermore, both *M. smegmatis* and *M. tuberculosis* possess the enzyme CO dehydrogenase to use CO as an energy source; it was recently demonstrated that *M. smegmatis* enhances its long-term survival by scavenging atmospheric CO when preferred organic energy sources are exhausted (9).

Mycobacteria are obligate aerobes, meaning they require a functioning aerobic respiratory chain to grow (17). As a result, the inhibitory effect of CO on the oxygen-binding heme groups of the terminal oxidases must be resisted for these bacteria to grow in the presence of CO. The mycobacterial respiratory chain is branched, with the final reduction of molecular oxygen mediated by one of two terminal oxidases, the cytochrome $bcc$-$aa_3$ supercomplex or the cytochrome $bd$ oxidase (18). The cytochrome $bcc$-$aa_3$ supercomplex is composed of components analogous to mitochondrial complex III and IV; primarily synthesized under optimal growth conditions (19), it is the more efficient of the two oxidases given that it acts as a proton pump (18). In contrast, cytochrome $bd$ oxidase is non-proton pumping and is therefore less efficient but is thought to have a higher $O_2$ affinity and is induced during hypoxia (20, 21). Cytochrome $aa_3$ oxidases are members of the heme-copper oxidase superfamily with binuclear heme-copper active sites that are highly susceptible to inhibition by ligands such as CO, nitric oxide (NO), cyanide ($CN^-$), and hydrogen sulfide ($H_2S$) (22). Cytochrome $bd$ oxidases are unrelated to heme-copper oxidases and utilize dual $b$ and $d$ hemes in their active site for $O_2$ reduction (21). In several bacteria, cytochrome $bd$ oxidase is important for resistance to oxidative and nitrosative stress, as well as to NO, CN, and $H_2S$, suggesting its active site is less susceptible to inhibition by non-$O_2$ ligands (23–25). In *Escherichia coli*, cytochrome $bd$-*I* oxidase is transcriptionally upregulated in cells grown in the presence of CO gas, and cells utilizing cytochrome $bd$-*I* as their sole terminal oxidase are resistant to inhibition by CORM-3, potentially due to resistance to CO released by this molecule (26, 27). These data point toward a role for cytochrome $bd$-*I* oxidase in CO resistance in *E. coli*, though it remains to be determined whether cytochrome $bd$ plays a similar role in resistance to CO in mycobacteria and in bacteria in general.

In addition to acting as an alternative energy source at low concentrations and a respiratory poison at high concentrations, CO has been shown to influence gene expression in mycobacteria via the two-component DosS/DosR system (8, 28). The sensor histidine kinase DosS is a hemoprotein that activates the transcriptional regulator DosR via phosphorylation in response to low oxygen, low redox state, or binding of ligands to its heme functional group (29). The DosS/DosR regulator has been most thoroughly

characterized in *M. tuberculosis*, which possesses an additional sensor kinase designated DosT that acts synergistically with DosS to modulate DosR function (30). In *M. tuberculosis*, the *dos* regulon contains at least 48 genes and contributes to survival during hypoxia-induced dormancy (31–34). While the Dos response plays a role in adaptation to hypoxia and in resistance to respiratory inhibition by NO, the physiological role of its activation in response to CO in *M. tuberculosis* has not been determined (8, 28, 31). Moreover, in *M. tuberculosis*, *dos* regulon activation in response to CO is much less pronounced than to NO and may potentially be a nonspecific effect relating to the inherent affinity of CO for the DosS heme functional groups (8). In *M. smegmatis*, the *dos* regulon plays a similar role to *M. tuberculosis* in preparing cells for hypoxia and is largely analogous, with the notable addition of the hydrogenases Hhy and Hyh, which support redox homeostasis (17, 35–37). The activation of the *dos* response by CO and a potential role in CO resistance in *M. smegmatis* has not previously been investigated.

In this study, we sought to determine the effect of CO on *M. smegmatis* throughout its growth cycle. We systematically compared the growth, proteome, and respiration of *M. smegmatis* in the presence and absence of CO. To understand the functional basis of CO adaptation, we used deletion mutants and CRISPR interference (CRISPRi) knockdowns of genes encoding terminal oxidases and DosR. Our results show that *M. smegmatis* uses cytochrome *bd* oxidase as a primary means of resisting inhibition of its respiratory chain by CO. Cytochrome *bd* oxidase is induced during growth in CO, and its oxidase activity is resistant to inhibition by CO. In addition, mutant and knockdown strains lacking cytochrome *bd* oxidase are impaired in their ability to grow in the presence of CO. These data provide the first direct evidence of the role of respiratory chain remodeling in resistance to CO in mycobacteria.

## RESULTS

***M. smegmatis* induces cytochrome *bd* oxidase and the *dos* regulon during growth in the presence of CO.** In order to determine the effect of CO on *M. smegmatis*, we compared the growth of wild-type *M. smegmatis* in glycerol-containing minimal medium in sealed vials under an atmosphere with either 20% $N_2$ or 20% CO. Under these conditions, growth of *M. smegmatis* was slower in the presence of 20% CO (Fig. 1A), though the exponential-phase growth rate and final growth yield were nearly identical under both conditions (Fig. 1B and C). The lag phase during growth in 20% CO was 1.3-fold longer than that in 20% $N_2$ (Fig. 1D), accounting for the difference in growth. This suggests that *M. smegmatis* is initially inhibited by CO but grows normally after adapting to the gas, likely through gene expression changes. In order to identify changes in the *M. smegmatis* proteome that may account for this adaptation to CO, we performed proteomic analysis on *M. smegmatis* cultures in the presence and absence of 20% CO during the mid-exponential phase.

Proteomic analysis showed that 37 proteins were differentially abundant in response to growth in the presence of 20% CO (Fig. 2A, Table S1), with 27 proteins more abundant (Fig. 2E) and 10 less abundant (Fig. 2F). The cytochrome *bd* oxidase subunits CydA and CydB were highly induced (24- and 4.8-fold) in response to growth in 20% CO, while levels of the cytochrome *bcc-aa$_3$* supercomplex (QcrCAB) were unaffected (Fig. 2A, Table S1). This suggests that in *M. smegmatis*, cytochrome *bd* oxidase production is induced to adapt to growth in CO and may help overcome respiratory inhibition by this gas. This is consistent with the established role of cytochrome *bd* oxidase in resistance to NO, CN, and $H_2S$ in *M. tuberculosis* and *E. coli* (23–25, 38), as well as its induction in response to CO and insensitivity to CORM-3 treatment in *E. coli* (26, 27). Fifteen of the most induced proteins belong to the *dos* regulon, representing a subset of this regulon, which includes the regulator of the pathway DosR (5.0-fold), universal stress protein family proteins (98- to 5.7-fold), and two proteins that protect against oxidative stress through the sequestration of flavins, Acg (1,007-fold) and Fsq (72-fold) (39, 40). The full *dos* regulon in *M. smegmatis* contains 49 proteins (17). Notable *dos*-regulated proteins not induced by CO include the hydrogenases Hhy and

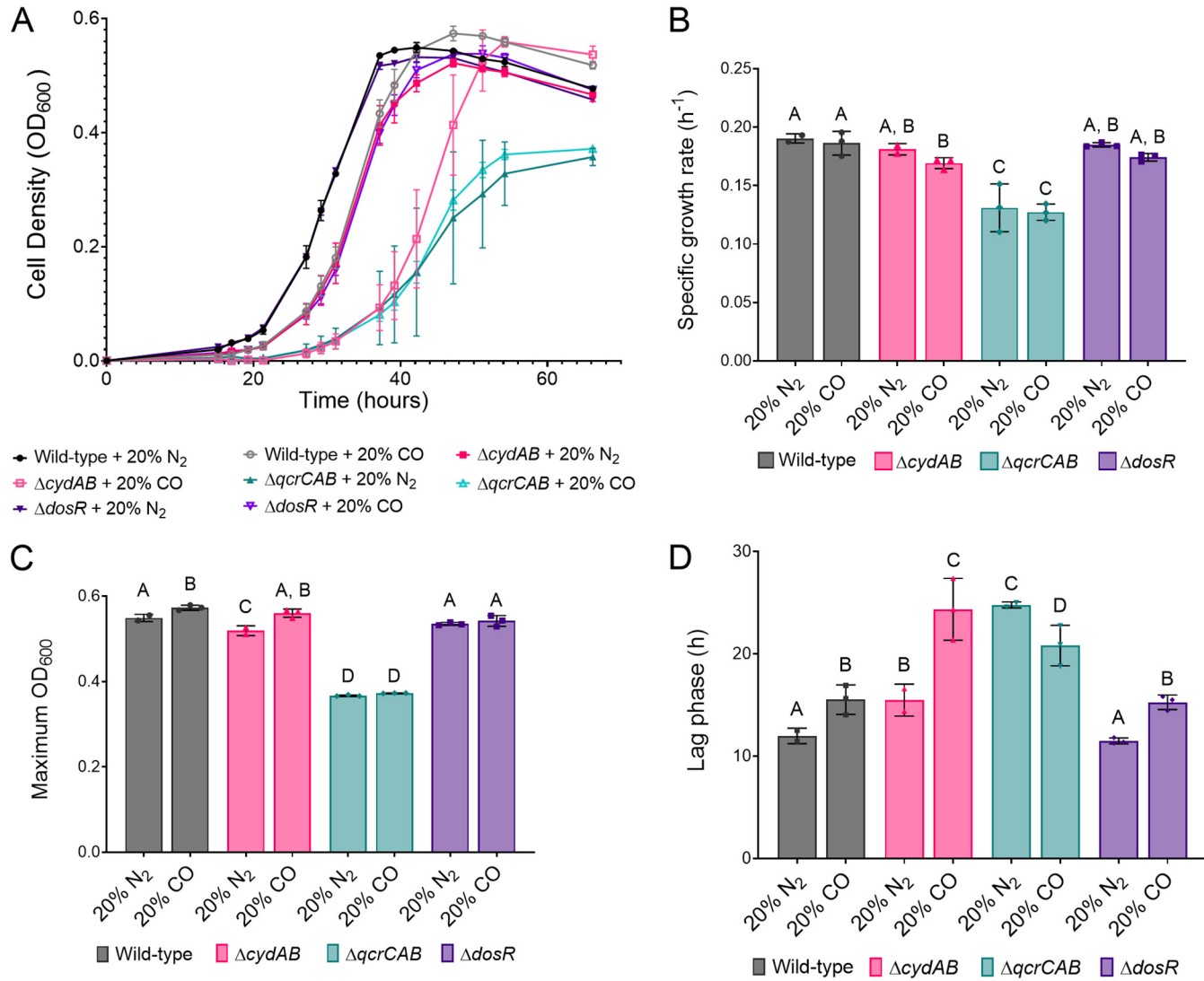

**FIG 1** Growth of *M. smegmatis* wild-type and terminal oxidase mutants in air supplemented with either 20% CO or 20% N₂. (A) Growth curves of *M. smegmatis* wild-type, terminal oxidase, and *dosR* mutants grown in sealed culture vials in the presence of air supplemented with 20% CO or 20% N₂. (B to D) The specific growth rate (B), maximum culture density (C), and length of lag phase (D) for each strain are also shown. Different letters above data bars (A, B, C, and D) designate significantly different values ($P$ value $< 0.05$, two-way analysis of variance [ANOVA]).

Hyh, which are important for the response to starvation and hypoxia, respectively (17, 41). It is unclear why the *dos* regulon in *M. smegmatis* is only partially induced by CO; this may reflect the interaction between sensor kinases and CO or cross talk between DosR and other regulatory mechanisms.

Fewer proteins decreased in abundance in response to CO exposure. The ESX-3 type VII secretion system component EccC3, known to be involved in the export of proteins important for iron acquisition (42), exhibited the largest decrease in abundance (15-fold). Correlating with this, the acyl-dehydrogenase MbtN involved in synthesis of the iron-binding siderophore mycobactin (43) was also less abundant (10-fold). The decreased abundance of proteins involved in iron acquisition suggests that *M. smegmatis* does not experience iron limitation during growth in CO, due to iron sequestration in Fe-carbonyl complexes. Iron limitation induced by this mechanism was proposed for *E. coli*, which was shown to transcriptionally upregulate iron-acquisition systems in response to CO (27).

The 37 proteins with differential abundance in response to CO comprise only 0.56% of the total 6,625 proteins predicted to be synthesized by *M. smegmatis* mc²155 (44).

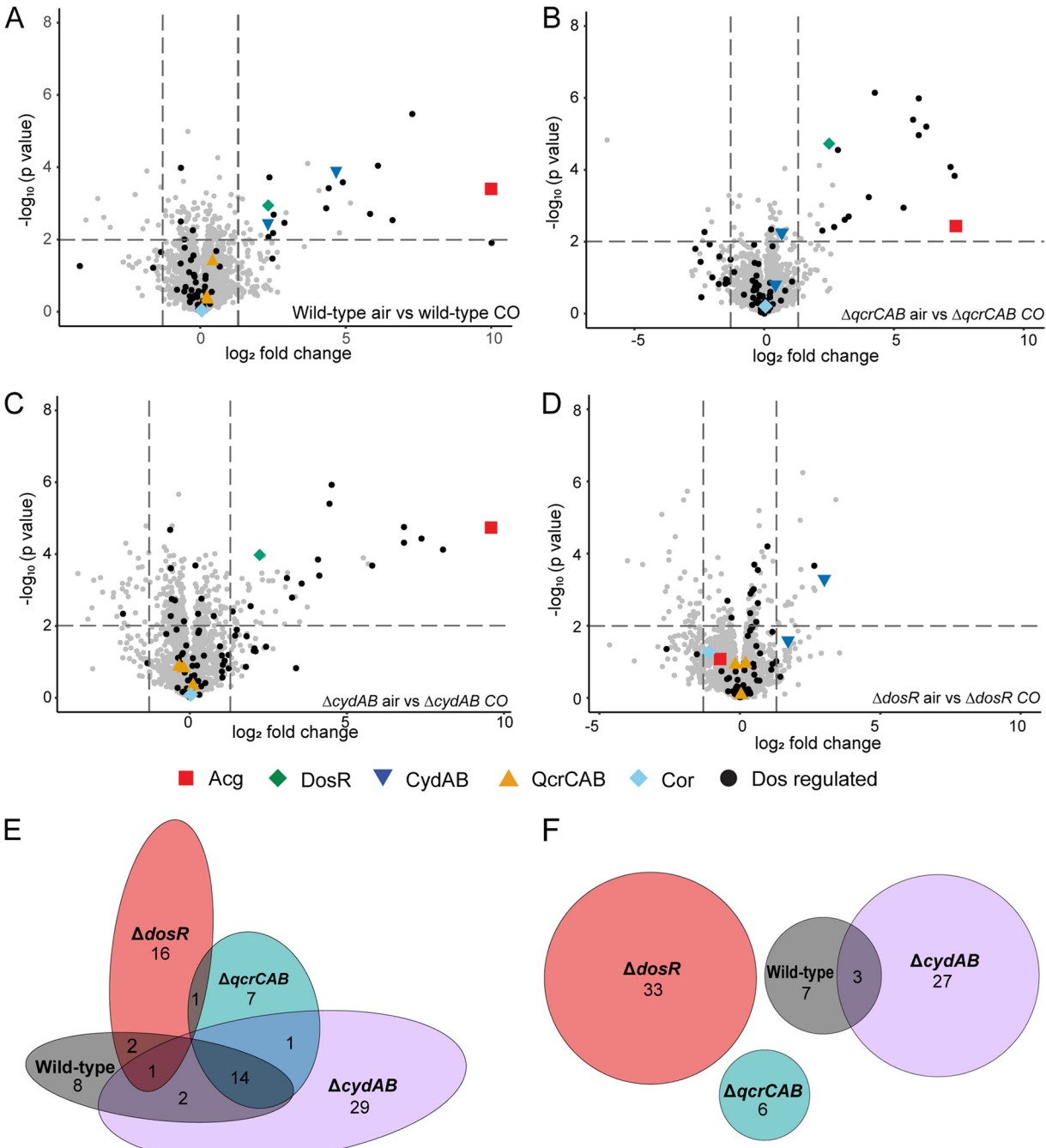

**FIG 2** Shotgun proteomic analysis of *M. smegmatis* wild-type and terminal oxidase mutants at mid-log phase grown in air in the presence and absence of 20% CO. (A to D) Volcano plots showing differential abundance of proteins in (A) wild-type, (B) Δ*qcrCAB*, (C) Δ*cydAB* and (D) Δ*dosR* strains harvested at mid-exponential phase ($OD_{600}$, ~0.3) grown in air with and without 20% CO. Each protein is represented by a single point, with *dos* regulon proteins represented by black dots and proteins of interest highlighted as per the legend. Dotted lines represent significance and fold change thresholds. (E and F) Venn diagrams showing proteins identified by the shotgun proteomics in multiple data sets with higher (E) and lower (F) abundances in wild-type, Δ*qcrCAB*, Δ*cydAB*, and Δ*dosR* mutant strains grown in air plus 20% CO.

This suggests that *M. smegmatis* has a high level of inherent resistance to CO toxicity, requiring relatively few changes to its proteome to cope with this gas. Among proteins that were not differentially abundant during growth in CO were Cor and CO dehydrogenase. Cor was reported to be the most important gene for CO resistance in *M. tuberculosis* (45) and is highly conserved in *M. smegmatis* (MSMEG_3645, 94% amino acid identity); while the exact function of this protein remains unresolved, these results

suggest Cor-mediated CO resistance is independent of induction by CO. The lack of induction of CO dehydrogenase in response to CO is consistent with previous findings that this enzyme is deployed to utilize trace quantities of CO as an energy source during persistence (9).

**Cytochrome *bd* oxidase, but not DosR, contributes to CO resistance.** Our proteomic analysis demonstrates that *M. smegmatis* induces cytochrome *bd* oxidase and members of the *dos* regulon in response to CO. To determine the role of the terminal oxidases and the *dos* regulon in resistance to CO, we systematically assessed the growth of *M. smegmatis* strains with previous constructed genetic deletions of the cytochrome *bcc-aa*$_3$ oxidase (ΔqcrCAB), cytochrome *bd* oxidase (ΔcydAB), and the regulator DosR (ΔdosR) (17, 35), as above, in the presence of 20% CO or 20% N$_2$. Growth of the ΔdosR strain was identical to that of the wild type in both the N$_2$- and CO-supplemented conditions (Fig. 1A). Thus, the probable inability of the ΔdosR strain to induce the *dos* regulon does not affect growth in CO and suggests that the partial induction of the *dos* regulon by CO is not an adaptive response to tolerate CO.

In contrast, there were significant differences in the growth characteristics of the terminal oxidase mutants in the presence and absence of CO. In line with previous observations (19, 46), in the absence of CO, the ΔqcrCAB mutant exhibited a longer lag phase, slower specific growth rate, and lower final growth yield than the wild type, whereas the ΔcydAB mutant exhibited only slightly slower growth than the wild type (Fig. 1A to D). The growth rate of the ΔqcrCAB strain did not differ in the presence or absence of CO, suggesting that cytochrome *bd* oxidase is inherently resistant to inhibition by CO (Fig. 1A and B). In contrast, the ΔcydAB strain grew markedly slower in the presence of CO compared to growth in the 20% N$_2$-amended control. Under these conditions, the slower growth is largely attributable to an increase in lag phase (Fig. 1D), as the specific growth rate of the ΔcydAB strain during exponential phase in CO was only slightly lower than that of the wild type (Fig. 1B). The increase in lag phase of the ΔcydAB strain in CO was 2.5-fold longer than for wild-type *M. smegmatis* (Fig. 1D). This is consistent with our proteomic analysis that shows cytochrome *bd* oxidase is induced in response to CO and demonstrates that this terminal oxidase is important for adaptation to growth in CO.

To validate the growth phenotypes associated with the ΔqcrCAB, ΔcydAB, and ΔdosR mutant strains, we utilized a CRISPRi strategy to independently repress the expression of *qcrC*, *cydA*, and *dosR* in a wild-type background strain. The anhydrotetracycline-inducible CRISPRi system utilized employs a catalytically inactive Cas9 enzyme in combination with a sequence-specific guide RNA (sgRNA) to bind to and repress transcription of a target gene. This system has been optimized for *M. smegmatis*, where it has been shown to achieve up to 200-fold repression of gene expression (47). Growth curves were conducted with the *qcrC*, *cydA*, and *dosR* knockdown strains, as well as a negative-control strain containing a scrambled nonspecific guide RNA. To ensure that sufficient levels of repression occurred throughout the experiment, a higher starting inoculum was utilized than in the experiments with the knockout strains (optical density at 600 nm [OD$_{600}$] of 0.01 versus 0.0005). The growth of *qcrC*, *cydA*, and *dosR* knockdown strains in the presence and absence of CO was consistent with that observed for the corresponding knockout strains (Fig. S1A). The higher starting OD precluded observation of the lag phase, though a lower exponential-phase specific growth rate was observed for the negative control, *qcrC*, and *dosR* knockdown strains in the presence of CO. The specific growth rate of the *cydA* knockdown in the presence of CO was lower (1.39-fold) than that of the scrambled control (1.13-fold) and *dosR* knockdown (1.21-fold) strains (Fig. S1B), which is consistent with the sensitivity of the cytochrome *bcc-aa*$_3$ complex to CO, observed in the ΔcydAB knockout strain. As observed for the ΔqcrCAB strain, the *qcrC* knockdown exhibited the same growth parameters in the presence and absence of the CO. No strains exhibited a significant difference in growth yield in the presence or absence of CO (Fig. S1C).

**Proteomic response to CO in the terminal oxidase and DosR deletion strains.** In order to determine whether additional proteome changes occur during the adaptation

of the *M. smegmatis* terminal oxidase mutants to CO, we performed proteomic analysis of the three mutant strains during exponential growth in the presence and absence of 20% CO. In the absence of CO, the Δ*qcrCAB* mutant significantly increased synthesis of the cytochrome *bd* oxidase subunits CydA (52-fold) and CydB (9.6-fold) compared to the wild type (Table S1), likely to compensate for the loss of the main terminal oxidase in line with previous reports (19). In the Δ*qcrCAB* strain, only 28 proteins significantly differed in abundance in the presence of CO (23 higher, 5 lower) (Fig. 2B; Table S1), including the same 15 proteins of the *dos* regulon upregulated in the wild type, as well as multiple hypothetical proteins (Fig. 2E; Table S1). The lack of additional significant changes to the proteome of this strain suggests that it is inherently resistant to CO. In the Δ*cydAB* strain, the abundance of a larger subset of 77 proteins changed in response to CO (47 higher, 30 lower) (Fig. 2C; Table S1). Most of these differentially regulated proteins are poorly characterized, making it difficult to assess their role in adaptation to CO. Other than the *dos* regulon, the induced proteins include enzymes from the thiamine biosynthetic pathway (ThiC, 8.2-fold; ThiD, 3.7-fold), the histidine biosynthetic pathway (HisD, 4.9-fold), and a NAD(P)$^+$ transhydrogenase (6.8-fold) (Table S1), suggesting considerable metabolic remodeling. However, we did not observe a significant increase in the production of potential alternative terminal reductases (e.g., putative nitrate or fumarate reductases). Overall, the additional proteome changes observed in the Δ*cydAB* mutant suggest a larger-scale response to cope with increased respiratory inhibition due to the loss of cytochrome *bd* oxidase.

To confirm the role of DosR in the partial activation of the *dos* regulon in response to CO, we performed proteomic analysis of the Δ*dosR* mutant in the presence and absence of CO (Fig. 2D). In the Δ*dosR* mutant, the *dos* regulon was not induced in the presence of CO (Table S1). An exception was MSMEG_2264, a putative endopeptidase associated with the hydrogenase Hhy, which is not consistently upregulated in other strains and thus is unlikely to be directly related to the *dos* response to CO. This demonstrates that DosR is responsible for the upregulation of *dos* regulon proteins in response to CO and that the observed increase in the level of these proteins does not mediate CO resistance. Multiple other proteins were modestly differentially regulated proteins in response to CO in the Δ*dosR* mutant (20 upregulated and 33 downregulated), the majority of which are not shared with other strains (Fig. 2D to F; Table S1).

**Cytochrome *bd* oxidase is resistant to CO inhibition in actively growing *M. smegmatis* cells.** Our finding that in *M. smegmatis*, cytochrome *bd* oxidase is important for optimal growth in the presence of CO and is induced under these conditions led us to hypothesize that this enzyme is inherently resistant to inhibition by CO. To test this hypothesis, O$_2$ consumption was monitored amperometrically in *M. smegmatis* wild-type, Δ*qcrCAB*, and Δ*cydAB* strains during the mid-log phase; cells were spiked with glycerol to simulate respiration, followed by treatment with CO-saturated buffer. Spiking of glycerol stimulated O$_2$ consumption in all strains (Fig. S2). Consistent with a high sensitivity of the cytochrome *bcc-aa*$_3$ complex to inhibition by CO, complete inhibition of O$_2$ consumption was observed in the Δ*cydAB* mutant upon addition of CO (Fig. 3A). Inhibition of the wild-type strain was significant but less pronounced than that for the Δ*cydAB* mutant, while the Δ*qcrCAB* mutant was not significantly inhibited by CO (Fig. 3A). To validate these data, we repeated the respirometry experiments using mid-log-phase cultures of the *M. smegmatis qcrC* and *cydA* knockdown strains. The sensitivity of these cultures to respiratory inhibition by CO was identical to that observed for their knockout equivalents. The scrambled control (equivalent to the wild type) exhibited partial inhibition of O$_2$ consumption in the presence of CO, while the *cydA* knockdown exhibited total inhibition, and the *qcrC* knockdown exhibited no inhibition due to CO (Fig. S3). These data confirm our hypothesis that *M. smegmatis* cytochrome *bd* oxidase is resistant to inhibition by CO, while the cytochrome *bcc-aa*$_3$ supercomplex is highly sensitive to inhibition by the gas.

Recently, we demonstrated that *M. smegmatis* can utilize the trace quantities of CO present in the atmosphere as an energy source during starvation, via the CO dehydrogenase (9). This study demonstrated that the addition of CO leads to enhanced O$_2$

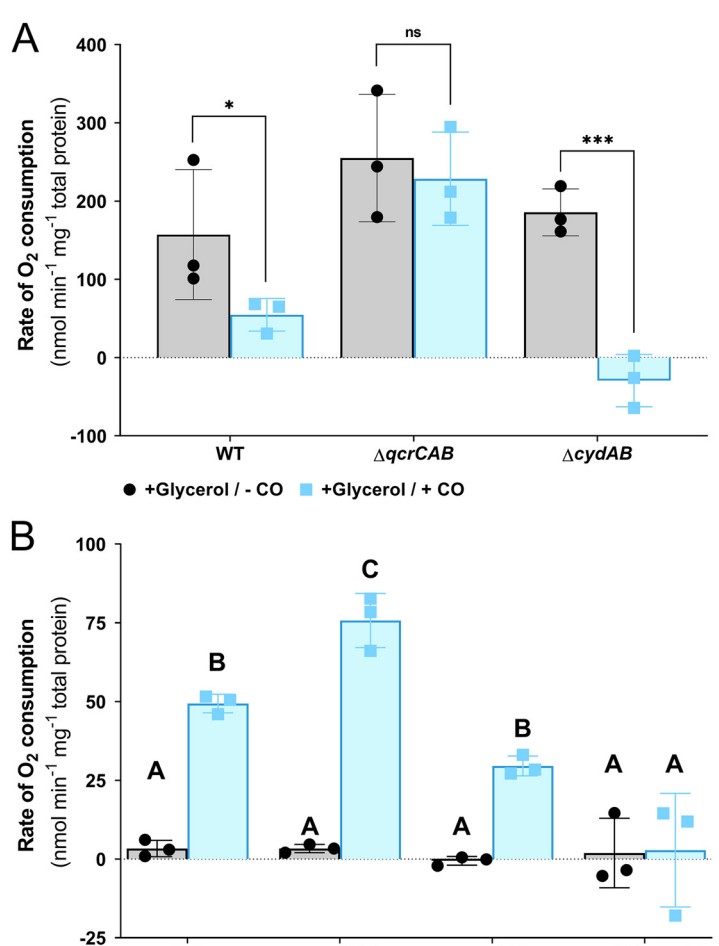

**FIG 3** Oxygen consumption of *M. smegmatis* terminal oxidases in the presence of CO. (A) Rate of $O_2$ consumption by *M. smegmatis* wild-type, Δ*qcrCAB*, and Δ*cydAB* mid-exponential, glycerol-spiked cultures in the presence and absence of CO. ns, nonsignificant; *, $P < 0.05$; ***, $P < 0.001$ (paired *t* test). (B) Rates of $O_2$ consumption of carbon-starved (3-days post-$OD_{max}$) *M. smegmatis* wild-type, Δ*qcrCAB*, and Δ*cydAB* cultures before and after spiking with CO. $O_2$ consumption was measured with an oxygen electrode. Azide is a compound that targets the cytochrome *bcc-aa3* oxidase and therefore acts as a negative control. Error bars represent the standard deviation of three biological replicates. Different letters above the data bars (A, B, and C) designate significantly different values ($P < 0.05$, two-way ANOVA).

consumption in *M. smegmatis*, suggesting that electrons derived from CO oxidation enter the respiratory chain, though the specific terminal oxidases involved in this process were not determined (9). To test this, we spiked carbon-limited (3 days post-$OD_{max}$) *M. smegmatis* wild-type, Δ*qcrCAB*, and Δ*cydAB* cultures with CO-saturated buffer and amperometrically monitored $O_2$ consumption. Upon addition of CO, all cultures consumed $O_2$, with consumption of the Δ*cydAB* culture being 1.7-fold less than that of the wild type (Fig. 3B). Conversely, $O_2$ consumption in the Δ*qcrCAB* culture was 1.5-fold higher than that in the wild type (Fig. 3B). Inhibition of cytochrome *bcc-aa3* through the addition of zinc azide in the Δ*cydAB* mutant completely abolished CO-dependent $O_2$ consumption (Fig. 3B).

These data demonstrate that electrons generated by CO oxidation by CO dehydrogenase can be donated to either terminal oxidase. Furthermore, the complete inhibition of $O_2$ consumption by zinc azide in the Δ*cydAB* mutant suggests that $O_2$-dependent CO oxidation is obligately coupled to the terminal oxidases of the respiratory chain. The higher rate of $O_2$ consumption in the Δ*qcrCAB* mutant may result from the

insensitivity of cytochrome *bd* to inhibition by CO. In the wild-type and Δ*cydAB* strains, it is likely that the addition of CO concurrently stimulates $O_2$ consumption by providing electrons to the electron transport chain and inhibits respiration through inhibition of cytochrome *bcc-aa*$_3$ oxidase. As cytochrome *bd* oxidase is insensitive to CO, only a stimulatory effect is observed in the Δ*qcrCAB* mutant. The respiratory stimulation observed upon addition of CO to the Δ*cydAB* strain during carbon limitation contrasts with the complete inhibition observed when CO is added to actively growing glycerol-stimulated cells (Fig. 3A and B).

## DISCUSSION

Previous investigations showed that *M. smegmatis*, and mycobacteria more generally, exhibit considerable resistance to CO (8, 9, 16). However, prior to our current work, no underlying mechanism for CO resistance in mycobacteria had been determined. Here, we show that induction of cytochrome *bd* oxidase is the key adaptive response for CO resistance in *M. smegmatis*, as the function of this oxidase is largely unaffected by high concentrations of CO. This finding adds to a growing body of evidence that cytochrome *bd* oxidases play a general role in the resistance of the bacterial respiratory chain to gaseous inhibitors (48). The resistance of cytochrome *bd* oxidase to CO in *M. smegmatis* is consistent with a previous report that an *E. coli* mutant possessing only cytochrome *bd-I* oxidase is resistant to inhibition by the CO-producing molecule CORM-3 (26). However, a recent study demonstrated that purified cytochrome *bd-I* and *bd-II* oxidases from *E. coli* are more sensitive to inhibition by gaseous CO than cytochrome *bo'* oxidase (49). The difference between these findings may result from the use of CORM-3 for CO delivery in the former study. CORM-3 is known to exert cytotoxic effects independent of CO, and so the resistance of cytochrome *bd-I* oxidase to this compound may involve multiple factors (12). As such, our current work confirms that a bacterial cytochrome *bd* oxidase displays inherent resistance to gaseous CO in a physiological context.

The proteomic analyses showed that surprisingly few proteins are differentially abundant in *M. smegmatis* during growth in a 20% CO atmosphere. Fifteen of the most induced proteins, including the only proteins consistently induced in both the wild-type and terminal oxidase mutant backgrounds, belong to the *dos* regulon of *M. smegmatis* (17). Our data show that these proteins are induced by DosR and that this induction is not required for adaptation to growth in CO. This further reduces the number of adaptive proteomic changes observed in response to CO in *M. smegmatis*. Excluding DosR-regulated proteins and cytochrome *bd* oxidase, only 21 proteins are differentially abundant during growth of wild-type *M. smegmatis* in CO, with most of these changes less than 5-fold compared to growth in air. This indicates that cytochrome *bd* oxidase plays a key role in CO resistance and that *M. smegmatis* is otherwise highly resistant to CO poisoning. It is probable that two mechanisms allow cytochrome *bd* oxidase to maintain respiratory function in the presence of CO: (i) increased transcription as indicated by the proteomics and (ii) rerouting of electrons to the cytochrome *bd* oxidase when cytochrome *bcc-aa*$_3$ complex is inhibited. Further studies of purified cytochrome *bcc-aa*$_3$ supercomplex and cytochrome *bd* oxidase would be useful to confirm the differential CO susceptibility of these enzymes and mechanisms of CO inhibition.

Previously, we showed that, in *M. smegmatis*, CO dehydrogenase is active during nonreplicative persistence (9). This provides *M. smegmatis* with the ability to utilize CO at atmospheric concentrations as an alternative energy source in the absence of organic substrates (9). Our current work shows that electrons derived from CO are donated to $O_2$ via either of the two terminal oxidases, obligately coupling CO dehydrogenase to the aerobic respiratory chain. The observation that cytochrome *bcc-aa*$_3$ oxidase can accept electrons from CO oxidation, even though it is inhibited by the gas, is seemingly paradoxical. However, CO is a competitive inhibitor of $O_2$ binding by heme-copper oxidases, and CO dehydrogenase in *M. smegmatis* is a high-affinity enzyme that operates at very low CO partial pressures (9, 22). This means that, at the

mSystems®

low concentrations of CO that are physiologically relevant for CO dehydrogenase activity in *M. smegmatis*, no significant inhibition of cytochrome *bcc-aa*$_3$ oxidase would be observed.

Despite the suggestion that CO has antibacterial potential against pathogenic mycobacteria and numerous other bacterial species (5, 13, 50), surprisingly little is known about the physiological and biochemical effects of gaseous CO on the bacterial cell. The uncertainty regarding the effects of CO on bacteria is confounded by the fact that much of the work testing the effects of CO has been performed using CORMs, which have antibiotic activity in addition to the effects of CO release (12). In this work, we have shown that *M. smegmatis* has a high level of resistance to CO that requires relatively few changes to its proteome. Further studies should test whether these findings made using *M. smegmatis* as a model system extend to pathogenic mycobacteria. If *M. tuberculosis* also uses respiratory remodeling to tolerate CO, it is unlikely that CO produced either by the host via heme oxygenase 1 (8) or delivered exogenously will exert a significant antibacterial effect on this pathogen.

## MATERIALS AND METHODS

***M. smegmatis* strains utilized in this study.** A full list of strains used in this study is available in Table S2. Wild-type *Mycobacterium smegmatis* mc$^2$155 (51) and the derived mutants ΔqcrCAB, ΔcydAB, and ΔdosR were a gift from Gregory Cook of the University of Otago (17, 35). The ΔqcrCAB and ΔcydAB strains contained markerless deletions, while the ΔdosR strain contained a hygromycin resistance cassette in place of *dosR*. The authenticity of wild-type and mutant strains was confirmed via PCR amplification of the deleted region (Fig. S4) using the primers shown in Table S3. Unless otherwise stated, all cultures were maintained in lysogeny broth supplemented with 0.05% (wt/vol) Tween 80 (LBT) or 1.5% LBT agar.

**CRISPRi knockdown strain construction.** CRISPRi knockdown strains of *cydA*, *qcrC*, and *dosR* were constructed as per the method described in Rock et al. (47). The kanamycin-selectable CRISPRi plasmid pLJR962 (Addgene plasmid no. 115162; https://www.addgene.org/115162/) utilized was a gift from Sarah Fortune (47). To achieve a knockdown of *cydA*, *qcrC*, and *dosR*, sgRNAs were designed targeting the nontemplate strand within the open reading frame of these genes, consisting of a 21-bp sequence immediately 5′ to a protospacer-adjacent motif (PAM) located within the template strand, as shown in Table S3. This 21-bp sequence was designated the forward oligonucleotide, and its complementary sequence, the reverse oligonucleotide, overhangs compatible with annealing with DNA cleaved with BsmBI were added to these oligonucleotides (Table S3). The forward and reverse oligonucleotides were synthesized (IDT, Australia) and annealed in a 50-μl reaction mixture containing 20 μM (each) oligonucleotides and T4 ligase buffer (New England Biolabs [NEB]); heating at 95°C for 5 min was followed by cooling to 25°C at a ramp rate of 0.1°C s$^{-1}$. The annealed oligonucleotides were inserted into pLJR962 via Golden Gate cloning as described previously (52). Briefly, a 10-μl reaction was set up containing T4 DNA ligase (200 U) and T4 ligase buffer (NEB), BsmBI (5 U; NEB), pLR962 (25 ng), and annealed oligonucleotides (1 μM). The reaction mixture was incubated for 30 cycles of digestion (42°C, 5 min) and ligation (16°C, 5 min). The resulting plasmids were propagated in *E. coli* DH5α and sequenced to confirm correct insertion of the sgRNA oligonucleotide and then transformed into *M. smegmatis* via electroporation and plated onto LBT agar plates supplemented with 20 mg ml$^{-1}$ kanamycin. Kanamycin-resistant colonies were selected and screened via PCR to confirm genomic integration of pLJR962 containing the desired sgRNA.

**Growth experiments.** For experiments to determine the effect of CO on the growth of *M. smegmatis*, cultures were grown in Hartmans de Bont minimal medium (53) supplemented with 0.05% (wt/vol) tyloxapol and 2.9 mM glycerol and incubated at 37°C on a rotary incubator at ~180 rpm. For the assessment of *M. smegmatis* mc$^2$155 wild-type, ΔqcrCAB, ΔcydAB, and ΔdosR knockout strains, two sets of triplicate cultures were grown in 30 ml medium in 120-ml sealed serum vials. After inoculation at 0.0005 OD$_{600}$, 18 ml of ambient air was removed from the culture headspace by syringe and replaced with 18 ml of CO (via 100% vol/vol CO gas cylinder, 99.999% pure) at 1 atm to give a final CO concentration of 20%. To account for the O$_2$ removed from the CO-treated vials, the headspace of the other triplicate was analogously substituted with N$_2$ (via 100% vol/vol N$_2$ cylinder, 99.999% pure) to a final concentration of 20%. Growth was monitored by measuring the optical density at 600 nm (OD$_{600}$) (1-cm cuvette; Eppendorf BioSpectrometer basic). When the OD$_{600}$ was above 1, cultures were diluted 10-fold before reading. Assessment of the effect of CO on the growth of *M. smegmatis* qcrC, cydA, and dosR knockdown strains was performed as for the knockout strains, but with a higher starting inoculation of 0.01 OD$_{600}$, a glycerol concentration of 5.8 mM, and the addition of 200 ng ml$^{-1}$ anhydrotetracycline to induce the CRISPRi system. In place of wild-type *M. smegmatis*, a strain transformed with pLJR962 containing a nontargeting scrambled sgRNA was used. Vials for the growth of knockdown strains were foil-wrapped to prevent degradation of the light-sensitive anhydrotetracycline. Growth rates for all experiments were calculated by fitting the exponential portion of the growth curves via nonlinear regression (least-squares fit) in GraphPad Prism 8.3.0 for Windows. Regression was performed individually for each replicate. Maximum OD$_{600}$ was defined as the highest single OD$_{600}$ measure taken for each replicate. For knockout cultures, lag phase was defined as the time postinoculation taken to reach 0.01 OD$_{600}$. Specific growth

rate, maximum $OD_{600}$, and lag phase were assessed for statistical significance via two-way ordinary ANOVA and Tukey's multiple-comparison tests performed in GraphPad Prism 8.3.0.

**Proteomic analysis.** For shotgun proteomic analysis, 30-ml cultures of *M. smegmatis* wild-type and $\Delta qcrCAB$, $\Delta cydAB$, and $\Delta dosR$ mutant strains were grown on Hartmans de Bont minimal medium (53) supplemented with 0.05% (wt/vol) tyloxapol and 5.8 mM glycerol in 120-ml serum vials sealed with rubber butyl stoppers. For each strain, two sets of triplicate cultures were prepared. After inoculation, one triplicate set was amended with 20% CO (via 100% vol/vol CO gas cylinder, 99.999% pure), while the other was not treated. Cells were harvested in the mid-exponential phase ($OD_{600}$, ~0.3) by centrifugation ($10,000 \times g$, 10 min, 4°C). They were subsequently washed in phosphate-buffered saline (PBS; 137 mM NaCl, 2.7 mM KCl, 10 mM $Na_2HPO_4$, and 2 mM $KH_2PO_4$, pH 7.4), recentrifuged, and resuspended in 100 mM Tris + 4% SDS at a weight:volume ratio of 1:4. The resultant suspension was then lysed by beat-beating with 0.1 mm zircon beads for five 30 s cycles. To denature proteins, lysates were boiled at 95°C for 10 min and then sonicated in a Bioruptor (Diagenode) using 20 cycles of "30 seconds on" followed by "30 seconds off." The lysates were clarified by centrifugation ($14,000 \times g$, 10 min). Protein concentration was confirmed using the bicinchoninic acid assay kit (Thermo Fisher Scientific) and normalized for downstream analyses. After removal of SDS by chloroform/methanol precipitation, the proteins were proteolytically digested with trypsin (Promega) and purified using OMIX C18 Mini-Bed tips (Agilent Technologies) prior to liquid chromatography tandem mass spectrometry (LC-MS/MS) analysis. Using a Dionex UltiMate 3000 RSLCnano system equipped with a Dionex UltiMate 3000 RS autosampler, the samples were loaded via an Acclaim PepMap 100 trap column (100 $\mu$m × 2 cm, nanoViper, $C_{18}$, 5 $\mu$m, 100 Å; Thermo Scientific) onto an Acclaim PepMap RSLC analytical column (75 $\mu$m × 50 cm, nanoViper, $C_{18}$, 2 $\mu$m, 100 Å; Thermo Scientific). The peptides were separated by increasing concentrations of buffer B (80% acetonitrile/0.1% formic acid) for 158 min and analyzed with an Orbitrap Fusion Tribrid mass spectrometer (Thermo Scientific) operated in data-dependent acquisition mode using in-house, label-free quantification (LFQ)-optimized parameters. Acquired .raw files were analyzed with MaxQuant 1.6.5.0 (54) to globally identify and quantify proteins pairwise across the different conditions. Experiments comparing wild-type, $\Delta qcrCAB$, and $\Delta cydAB$ strains with and without CO were conducted separately from experiments comparing wild-type and $\Delta dosR$ strains with and without CO. After filtering, 3,119 proteins were identified in the terminal oxidase proteomic experiment, and 2,800 in the $\Delta dosR$ experiment. Statistical significance was determined in Perseus (55). Differentially regulated proteins were identified by assigning a fold change cutoff $>\pm 1.3$ and a -log($P$ value) of $> 2$. Volcano plots were generated from these data using R studio (56) with the ggplot2 package (57). Venn diagrams were generated with the Eulerr package (58).

**Respirometry measurements.** For respirometry measurements of knockout strains, 30-ml cultures of wild-type, $\Delta qcrCAB$, and $\Delta cydAB$ *M. smegmatis* were grown on Hartmans de Bont minimal medium (53) supplemented with 0.05% (wt/vol) tyloxapol and 5.8 mM glycerol to the mid-exponential ($OD_{600}$, 0.3) or mid-stationary phase (72 h post $OD_{max}$, ~0.9) in 125-ml aerated conical flasks. Rates of $O_2$ consumption were measured using a Unisense $O_2$ microsensor in 1.1-ml microrespiration assay chambers that were stirred at 250 rpm at room temperature. Prior to measurement, the electrode was polarized at $-800$ mV for 1 h with a Unisense multimeter and calibrated with $O_2$ standards of known concentration. Gas-saturated phosphate-buffered saline (PBS; 137 mM NaCl, 2.7 mM KCl, 10 mM $Na_2HPO_4$, 2 mM $KH_2PO_4$, pH 7.4) was prepared by bubbling PBS with 100% (vol/vol) of either $O_2$ or CO for 5 min. To assess the effect of CO on the respiration of growing cells, 0.9- ml mid-exponential-phase cultures of either *M. smegmatis* wild-type, $\Delta qcrCAB$, or $\Delta cydAB$ strains and 0.1 ml $O_2$-saturated PBS were loaded into respiration chambers, and the baseline rate of $O_2$ consumption was measured. Following this, glycerol (3.5 mM final concentration) and 0.1 ml CO-saturated PBS were sequentially amended into the chamber, and $O_2$ consumption was measured before and after addition of CO. To determine the effect of CO in carbon-starved cells, initial oxygen consumption was measured in assay chambers sequentially amended with mid-stationary-phase *M. smegmatis* cell suspensions (0.9 ml) and $O_2$-saturated PBS (0.1 ml). After initial measurements, 0.1 ml of CO-saturated PBS was added to the assay mixture. Additionally, $O_2$ consumption was measured in a $\Delta cydAB$ strain treated with 250 $\mu$M zinc azide. Respirometry experiments were performed on the wild-type, *cydA*, and *qcrC* CRISPRi knockdown strains during the exponential phase as for the knockout strains, with the addition of 300 ng ml$^{-1}$ of the inducer anhydrotetracycline to the culture medium immediately before inoculation. Vials were foil-wrapped to prevent anhydrotetracycline degradation. Changes in $O_2$ concentrations were recorded using Unisense Logger software (Unisense, Denmark). After a linear change in $O_2$ concentration was observed, rates of consumption were calculated over a period of 20 s and normalized against total protein concentration. Statistical significance for all experiments was assessed via two-way analysis of variance (ANOVA) with Tukey's multiple-comparison test or paired $t$ tests as indicated in the figure legends calculated using GraphPad Prism 8.3.0.

**Data availability.** All raw proteomic data have been deposited at PRIDE with the data set identifier PXD018382.

## SUPPLEMENTAL MATERIAL

Supplemental material is available online only.

**FIG S1**, TIF file, 0.7 MB.

**FIG S2**, TIF file, 0.1 MB.

**FIG S3**, TIF file, 0.2 MB.

**FIG S4**, TIF file, 0.5 MB.
**TABLE S1**, XLSX file, 0.3 MB.
**TABLE S2**, DOCX file, 0.01 MB.
**TABLE S3**, DOCX file, 0.01 MB.

## ACKNOWLEDGMENTS

This work was supported by an ARC Discovery grant (DP200103074; awarded to C.G. and R.G.), an ARC DECRA fellowship (DE170100310; awarded to C.G.), an NHMRC EL2 fellowship (APP1178715; awarded to C.G.), an Australian Government Research Training Program stipend scholarship (awarded to K.B.), and a Monash University doctoral scholarship (awarded to P.R.F.C.).

We thank Gregory Cook and Matthew McNeil for providing the ΔqcrCAB, ΔcydAB, and ΔdosR mutants.

C.G. and R.G. conceived and supervised the study. C.G., K.B., P.R.F.C., and R.G. designed experiments. Different authors were responsible for growth experiments (K.B., A.K., R.G.), knockdown construction (P.R.F.C.), proteomic analysis (C.H., R.B.S., K.B.), and respirometry experiments (P.R.F.C.). K.B., P.R.F.C., C.G., R.G., and C. H. analyzed data. R.G., K.B., P.R.F.C., and C.G. wrote the paper with input from all authors.

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
