## [Reviewer comments · mSystems]

Mycobacteria tolerate carbon monoxide by remodeling their respiratory chain

Katherine Bayly, Paul Cordero, Ashleigh Kropp, Cheng Huang, Ralf Schittenhelm, Rhys Grinter, and Chris Greening

Corresponding Author(s): Chris Greening, Monash University

Review Timeline:

Submission Date:	December 7, 2020
Editorial Decision:	February 4, 2021
Revision Received:	February 12, 2021
Accepted:	April 14, 2021

Editor: Cynthia Collins

Reviewer(s): Disclosure of reviewer identity is with reference to reviewer comments included in decision letter(s). The following individuals involved in review of your submission have agreed to reveal their identity: Tiago Beites (Reviewer #4)

Transaction Report:

DOI: <https://doi.org/10.1128/mSystems.01292-20>

Response to editor comments

Re: mSystems00311-20 (Mycobacteria tolerate carbon monoxide by remodelling their respiratory chain)

Dear Prof. Chris Greening:

I have received the reviews of your manuscript. While your paper addresses an interesting question, the reviewers stated several concerns about your study and did not recommend publication in mSystems. In particular, please note that further descriptions of the genetic strains, additional proteomic analyses, and more detailed CO measurements are suggested. The manuscript may also benefit from placing this work more in context with previous cytochrome bd studies, and the authors claims may be further supported by purification and characterization of the enzyme.

As you know, at mSystems we are committed to making rapid final decisions. Because it appears that addressing the reviewers' concerns will require a significant amount of additional work that would delay the ultimate outcome, my decision at this time is to reject the manuscript.

If you feel that you wish to address the criticisms of the reviewers, you may submit a revised manuscript to mSystems as a new submission, which will be assigned a new manuscript number and receipt date. Please note the previous manuscript number and my name in the cover letter. Provide point-by-point responses to the issues raised by the reviewers in a file named "Response to Reviewers," not in your cover letter. Upload a compare copy of the manuscript (without figures) as a "Marked-Up Manuscript" file. In the response file, specify with page and line numbers where the revisions have been made in the marked-up manuscript.

I am sorry to convey a negative decision on this occasion, but I hope that the enclosed reviews are useful.

Sincerely,

Cynthia Collins

Editor, mSystems

We thank the editor for handling the manuscript and providing the option to resubmit the manuscript. Based on the reviewers' major suggestions, over the course of the last six months, we have performed a series of additional experiments that provide support for our major findings and add to the rigor of the manuscript. We have also made major revisions to the

text to better support, explain, and contextualize findings. The main changes are outlined below:

- 1. Reproduction of the CO-induced growth and respiration phenotypes using CRISPRi knockdowns of the *cydA*, *qcrC*, and *dosR* genes. This confirms that the effects we observed in the deletion strains were caused by the loss of the targeted genes. This has resulted in the addition of new growth data in Figure S1, respirometry data in Figure S3, and construction details in Table S2.*
- 2. Comparative proteomic analysis of the wild-type vs *dosR* mutant. This confirms that the CO-induced *dos* regulon proteins are indeed regulated by *DosR*. This has resulted in a new volcano plot in Figure 2 and additional comparisons in the Venn diagrams.*
- 3. Description of the source of the knockout strains and confirmation of their authenticity by PCR. This has resulted in the addition of Table S2 and Figure S4.*
- 4. Considerable revision of the manuscript, including a better comparison with the *E. coli* cytochrome *bd* literature, better emphasizing of where the novelty of the manuscript lies, and mentioning that future studies on purified protein complexes would also be an important (but highly challenging) next step.*

These changes address the reviewers' comments and ensure our findings are fully substantiated. We thank you for considering our revised submission of this manuscript.

Response to reviewer comments

Reviewer #1 (Comments for the Author):

This manuscript by Bayly et al. titled "Mycobacteria tolerate carbon monoxide by remodeling their respiratory chain" addresses an important question in microbiology, namely, why some bacteria are more resistant to carbon monoxide (CO) than others. Specifically, CO can be toxic to some bacteria, such as *E. coli*, however, *M. tuberculosis* (Mtb) and *M. smegmatis* (Msm) are highly resistant to CO toxicity. To better understand how mycobacteria can resist the toxicity of CO, Bayly et al. utilized Msm to characterize the Msm response to CO through proteomic profiling and growth analysis. The main findings are that Msm are resistant to CO toxicity and that cytochrome bd oxidase and proteins in the Dos regulon are upregulated in response to 20% CO. The authors used genetic deletion strains of cytochrome bd oxidase, cytochrome bcc-aa3 oxidase, and DosR and analyzed their growth in the presence of CO. The authors show that Msm lacking the cytochrome bd oxidase has reduced growth in the presence of CO, suggesting its role in CO resistance. This is an interesting manuscript, but the data are incomplete, and there are concerns about aspects of the methodology.

We thank the reviewer for the interest in our work and their constructive comments regarding the manuscript. We have thoroughly revised the manuscript, taking their comments into account, and performed additional experimental work.

Major concerns:

The primary concern regards the Msm genetics. The authors do not provide any data regarding the knockout strains used in this study. Did the authors create them themselves or obtain them from another lab?

We apologize for not providing these important details in the original manuscript. The strains were a gift from Prof. Gregory Cook at the University of Otago and were created in his laboratory. We have included details on the origin of the strains, as well as how the deletion was performed (i.e. gene deletion or substitution) in the revised manuscript. These details are provided in Table S2 and the following lines in the methods section:

L363-368: A full list of strains used in this study is available in Table S2. Wild-type *Mycobacterium smegmatis* mc²155 (1) and derived mutants $\Delta qcrCAB$, $\Delta cydAB$, and $\Delta dosR$ were a gift from Prof. Gregory Cook of the University of Otago (2, 3). The $\Delta qcrCAB$ and $\Delta cydAB$ strains contained markerless deletions, while the $\Delta dosR$ strain contained a hygromycin resistance cassette in place of *dosR*. The authenticity of wildtype and mutant strains was confirmed via PCR amplification of the deleted region (Figure S4), using primers shown in Table S3.

If the strains were generated in the lab, the authors need to provide evidence of genomic loss, either by PCR or Southern blotting or Western blotting if antibodies for the proteins exist, and the methods section needs to clearly explain how the mutants were made (primers, cloning strategy, recombination strategy, etc.).

We have performed PCR to confirm that these strains contain the described gene deletion. These data are presented in Figure S4 of the revised manuscript.

Furthermore, as the study relies heavily on use of genetic knockouts, complementation of each mutant is needed in order to have confidence in the results as there could be polar effects or off-site mutations (depending on the mechanism used to knockout the genetic loci). Thus, experiments with Msm mutants should be done with complemented strains. These data are essential (i.e. evidence of KO and complementation).

*We agree with the reviewer that additional evidence directly supporting the role of the terminal oxidase and dosR gene deletions in the phenotypes we observe will greatly strengthen this work. However, complementation of the terminal oxidase knockouts would likely be very challenging given it would require the addition of two or three genes to replace the deleted structural genes (i.e. cydAB, qcrCAB); in our experience, good expression and synthesis of multiple genes 'in trans' is highly challenging to achieve in *M. smegmatis*, especially for membrane-bound metalloenzymes. As such, to address this point in the revised manuscript, we have elected to utilize a CRISPRi knockdown strategy to provide an independent line of evidence confirming the phenotype associated with the gene deletions.*

We have constructed CRISPRi knockdown strains for cydA, qcrC, dosR, and a negative control strain containing a scrambled guide RNA. All of these strains were constructed concurrently from the same wild-type background strain. Using these strains, we have repeated the growth experiments with and without CO and have observed phenotypes consistent with those of the knockout strains. Namely, the scrambled control, cydA, and dosR knockdown all exhibited slower growth in the presence of CO. The qcrC knockdown grew slower than the other strains, but its growth was not further affected by the presence of CO. Further, the growth of the cydA knockdown was initially slower in the presence of CO compared to the scrambled control or dosR knockdown. It should be noted that we used a higher starting OD (0.01 vs 0.0005), which was required due to the instability of the CRISPRi inducer (anhydrotetracycline), and hence couldn't observe effects on lag phase. These new data are provided in Figure S1.

We have also repeated the respirometry experiments assessing the inhibitory effect of CO on O₂ consumption by the terminal oxidases using the CRISPRi strains. These experiments, conducted with log-phase cultures, show inhibition profiles for the scrambled, cydA, and qcrC strains that are identical to those observed for the knockout strains. Namely partial inhibition of O₂ consumption due to CO for the scrambled control, total inhibition of the cydA knockdown, and no inhibition of the qcrC strain. These new data are provided in Figure S1.

*As with the experiments conducted with the knockout strains, these data indicate that in *M. smegmatis* dosR is not required for adaptation to growth in the presence of CO, that cytochrome bd is resistant to inhibition by CO, and that cytochrome bcc-aa₃ is highly susceptible to inhibition by CO. We strongly feel that the two independent lines of evidence now presented are sufficient to support the conclusions drawn in our manuscript.*

Another major concern comes from the proteomic analysis. Some strains were left out, such as the DosR KO strain. Though it is unlikely, there could be a different response regulator

signaling CO in Msm, and as such, the proteomic response of the DosR KO should be performed.

This is an excellent suggestion. We have now performed proteomic analysis on the DosR knockout strain grown in the presence and absence of CO. In the absence of DosR, the dos regulon proteins with increased abundance in the presence of CO are not induced. This demonstrates that DosR is indeed required for the partial induction of the dos regulon in response to CO, but that these changes do not affect CO tolerance. These new data are provided in Figure 2D.

Minor concerns:

The organization of the figures was not ideal and could be improved to more clearly demonstrate the data. The fact that figure 1 is a result of the data from figure 2 is counter-intuitive.

We understand the reviewer's point here. However, by using this layout we are presenting the related datasets produced in the manuscript in the same figure for easy comparison by the reader. While splitting these data into separate figures would allow the figures to directly correspond to the text, it would make a comparison of the relevant data more difficult. As such, after much consideration, we have elected to keep the figure format the same as the original submission.

Statistical analysis was not described at all in the methods section, nor was it clear in the figure legends. In Fig. 1 and 3, letters (A, B, C...) were used to describe a statistical result but wasn't explained.

We have included a description of the statistical analysis performed in a revised version of the manuscript in the relevant sections of the methods as quoted below, and have added a description of the notation used in the figure legends of Figures 1, 3, S1, and S3.

L417-419 (growth): "Specific growth rate, maximum OD₆₀₀, and lag phase were assessed for statistical significance via two-way ordinary ANOVA and Tukey's multiple comparisons tests performed in GraphPad Prism 8.3.0."

L449-451 (proteomics): "Statistical significance was determined in Perseus (4). Differentially regulated proteins were identified by assigning a fold-change cut-off of $> +/-1.3$ and a $-\log(p\text{-value}) > 2$."

L481-483 (respirometry): "Statistical significance for all experiments was assessed via two-way ANOVA with Tukey's multiple comparison test or paired t-tests as indicated in figure legends calculated using GraphPad Prism 8.3.0."

The authors used only Msm in this work. However, in the discussion, they extrapolate their interpretation to Mtb and "other pathogenic members of the genus". Since all of the experiments here were done with the environmental species, the discussion should be more conservative.

In the original and revised manuscript, we aimed to be conservative with the language used.

Notably, we stated in the original manuscript 'if these findings extend to pathogenic mycobacteria'. In the revised manuscript, we have clarified throughout that the study focused on M. smegmatis and emphasized in the final sentences of the discussion that further work is needed to study the CO tolerance of M. tuberculosis:

L355-359: "Further studies should test whether these findings made using *M. smegmatis* as a model system extend to pathogenic mycobacteria. If *M. tuberculosis* also uses respiratory remodeling to tolerate CO, it is unlikely that CO produced either by the host via heme oxygenase 1 (5) or delivered exogenously will exert a significant antibacterial effect on this pathogen."

It should be noted that there are notable similarities between the response of Msm and Mtb to CO. Both grow in the presence of high concentrations of CO and induce the dos regulon in response to CO, as shown in studies by Kumar et. al. (6) and Shiloh et. al. (5). Additionally, Msm and Mtb have a similar respiratory chain structure, terminating in either the cytochrome bcc-aa₃ or cytochrome bd complexes. We, therefore, think it's important to note that if our findings translate to pathogenic bacteria, then it will have important ramifications for the utility of CO as an antimycobacterial compound.

Reviewer #3 (Comments for the Author):

Bayly et al describe the effects of CO on *M. smegmatis* and show that the organism can grow in its presence. Using a proteomic approach, they show that components of cytochrome bd and the Dos sensory system are the major protein species that are elevated when cells are cultured in the presence of CO gas dissolved in the growth medium. Using knockout mutants of the two major oxidases, they demonstrate that cytochrome bd confers CO resistance in *M. smegmatis*. Oxygen consumption in the presence of CO is eliminated in a cytochrome bd mutant. Finally, and most controversially, they develop a previous finding that *M. smegmatis* can oxidise CO. In a cytochrome bd mutant, this CO-stimulated rate of respiration is 1.7-fold lower than in the wild-type. This is not startling. It appears that the work is carefully done and well described. However, the main finding - that cytochrome bd confers CO resistance to *M. smegmatis* - is not surprising to anybody in the field of bacterial oxidases or CO. Similar studies have been reported before, especially in *E. coli*. The approach used here is proteomic not transcriptomic, but the findings are broadly the same.

We thank the reviewer for reviewing the manuscript and providing constructive comments and suggestions regarding our work.

*We agree with the reviewer that our findings are not startling, though they are important and well-supported. Considerable work has indeed been performed in *E. coli* assessing the role of cytochrome bd-I in resistance to CO. These papers are cited and discussed in our manuscript, and the consideration of them is refined and expanded in the revised manuscript (e.g. additions to L94-98, L314-323). Despite this, there are two major reasons why this study is warranted:*

1. It is pertinent to discuss CO tolerance specifically in mycobacteria. CO is highly important to study in this genus from a medical perspective (e.g. given *M. tuberculosis* is exposed to high CO levels through heme oxygenase 1) and environmental reasons (i.e. due to the discovery that soil mycobacteria are major sinks of CO and other trace gases). In addition to being highly medically and environmentally important, this genus is extremely phylogenetically divergent from *E. coli* and has a distinct physiology from *E. coli* as obligate aerobes rather than facultative anaerobe. In addition, mycobacteria have a distinct organization of respiratory chain complexes with the presence of a cytochrome *bcc-aa3* supercomplex that seems to be the main target of CO poisoning. Furthermore, the *CydA* and *CydB* subunits of cytochrome *bd* from *M. smegmatis* share a similarly low level of sequence identity (around 30-35%) with cytochrome *bd-I* and *bd-II* from *E. coli*. Thus, it cannot be guaranteed that principles discovered in *E. coli* would necessarily extend to mycobacteria. Consistently, we made both concordant and divergent findings with the *E. coli* literature.

2. It is difficult to predict how bacteria respond to CO due to conflicting literature in the area. For example, a study by Wareham et. al. (7) shows that in *E. coli* cytochrome *bd-I* is transcriptionally upregulated in the presence of CO, though this study does not investigate the physiological relevance of this upregulation. Likewise, a study by Jesse et. al. (8) shows that CO gas inhibits respiration of wild-type *E. coli*, and that cytochrome *bd-I* is resistant to inhibition by the CO releasing molecule CORM-3. While these studies are well-conducted and these data are strongly suggestive of a role for cytochrome *bd-I* in CO resistance in *E. coli*, they fall short of proof in our opinion because of the known non-CO related effects of CORM-3, well-illustrated in a study by Southam et. al. (9). Ambiguity regarding the resistance of *E. coli* cytochrome *bd-I* to CO has also been created by a study by Forte et. al. that shows that purified cytochrome *bd-I* from *E. coli* is more sensitive to inhibition by CO than the heme-copper oxidase cytochrome *bo'* (10). It is also notable that, while the available evidence does support a role for cytochrome *bd-I* in CO resistance, cytochrome *bd-II* was not upregulated in the study by Wareham et. al. (7) in response to CO. As such, we feel our paper is important and timely, as it goes some way to resolving this ambiguity by conclusively demonstrating (in *M. smegmatis* at least) that cytochrome *bd* oxidase in whole cells is resistant to inhibition by CO gas and that this resistance is relevant to the physiological response to this gas. We do not feel it is reasonable to infer that cytochrome *bd* from *M. smegmatis* will be resistant to CO solely based on the behavior of cytochrome *bd-I* from *E. coli*. This makes our experimental data showing that this is the case an important discovery.

The added element - that *M. smegmatis* can oxidize CO - is not persuasive (see below).

I would refer the reviewer to our previous study by Cordero et. al. (11) that conclusively demonstrates that *M. smegmatis* does oxidize CO, that this is dependent on the enzyme CO-dehydrogenase, and that this oxidation enhances survival during starvation. In this work we are not seeking to prove that *M. smegmatis* oxidizes CO, given this is already established, but rather we investigate how the electrons produced by this process are utilized by the terminal oxidases of the electron transport chain.

Detailed comments:

1. More attention should be paid to the concentrations of CO used in each experiment. Was this measured?

The reviewer is correct that we did not measure CO concentrations used in each experiment. However, we very carefully added defined amounts of excess CO into the CO treatment experiments, and as a laboratory, we specialize in reduced gas work. As outlined in the methods (lines 399-404, and 424-426) all bacterial cultures assessing the effect of CO on growth were sealed in 120 ml serum vials (containing 30 ml of culture) and amended with a headspace containing 20% CO gas. As outlined on lines 462-472, for the respirometry experiments 0.1 ml of CO saturated buffer was added to a 1 ml chamber volume containing 0.9 ml bacterial culture + 0.1 ml O₂ saturated buffer. While we agree in retrospect that the precise concentrations of CO gas in these experiments would have been desirable to measure, as we aimed to perform comparative analysis between strains using elevated CO concentrations, we do not consider this necessary to confirm our core conclusions.

2. The claim that it is not known whether cytochrome bd plays a role in CO resistance (line 95) is not justified: see ref 44.

In the revised manuscript, we have ensured we have avoided claims of primacy. As outlined above, ref 44 (Jesse et. al.) shows that cytochrome bd-I is resistant to inhibition by CORM-3 rather than CO gas. It is well established that CORM-3 has effects on cellular physiology outside of CO gas release, leaving some doubt whether the observed respiratory inhibition caused by CORM-3 is solely due to CO. This in turn leaves some doubt whether bd-I is resisting inhibition by CO or some other CORM-3 related mechanism. However, we agree that it is likely that the observed effect is likely related to CO. The Forte et al. study also indicates surprising CO sensitivity of purified cytochrome bd-I. In light of this, we have revised our manuscript to better reflect the E. coli work and the novelty of the present study, L94-98, L314-323, and the revised significance statement.

3. Lines 166-169 contrast the E. coli transcriptomics with the present proteomics. These results are not 'in contrast' with each other at all. It is generally the case that transcriptional effects are greater than effects on protein levels.

We agree that it's not appropriate to directly compare transcriptional changes with proteomic changes, as transcriptional effects are generally greater than the resulting change to the proteome. We feel that the large difference in magnitude of the change between the proteome of M. smegmatis and the transcriptome of E. coli is indicative of differences in the physiological response of these bacteria to CO. However, this direct comparison is not required for the point we wish to make in the paper (i.e. M. smegmatis requires relatively few changes in its proteome to grow in the presence of CO) so we have removed it from the revised manuscript.

4. It is unfortunate that more description of Cor is not provided. One of the more interesting outcomes is the finding that this putative CO resistance protein is not increased in level.

We agree that the fact that Cor is not induced in response to CO is interesting. While we dedicate six lines to discussing Cor, we couldn't reasonably speculate further on why this is given it remains to be established how Cor mediates CO resistance.

5. In Fig 3B, there are probably overlapping effects of CO on the CO oxidation rate per se and the ability of the oxidase to catalyse the resulting electron transfer.

We agree with the reviewer on this point and have addressed this with the following statement:

L297-301: "The higher rate of O₂ consumption in the $\Delta qcrCAB$ mutant may result from the insensitivity of cytochrome *bd* to inhibition by CO. In the wild-type and $\Delta cydAB$ strains, it is likely that the addition of CO concurrently stimulates O₂ consumption by providing electrons to the electron transport chain and inhibits respiration through inhibition of cytochrome *bcc-aa₃* oxidase."

6. Use of 250 micromolar azide is a pretty blunt tool. To unequivocally determine the CO resistance of cytochrome *bd*, studies with purified oxidase are needed.

*This experiment aimed to show that O₂ consumption of the $\Delta cydAB$ strain in response to CO requires a functional cytochrome *bcc-aa₃* complex, and thus that CO oxidation is coupled to the respiratory chain. Zinc azide is a known inhibitor of the cytochrome *bcc-aa₃* complex, so while it also likely has other cellular targets, the complete abolition of CO stimulated O₂ consumption when zinc azide is added is at least indicative that this is the case. This experiment was not intended to address the resistance of cytochrome *bd* to CO, which we already demonstrated in the growth curve and respirometry experiments with log-phase cells.*

*As discussed above, the study by Forte et. al. investigating the CO resistance of purified cytochrome *bd* oxidases from *E. coli* found that both *bd-I* and *bd-II* are more susceptible to inhibition by CO than cytochrome *bo'*. This conflicts with the aforementioned physiological data indicating that *bd-I* is important for CO resistance in *E. coli*. While the biochemical characterization of purified respiratory complexes is no doubt of considerable value, we feel that these need to be treated with caution as they do not consider the properties of the complex in its physiological context, and are probably inappropriate for unequivocally determining the physiological CO resistance. We have nevertheless added that further studies on the purified complexes would be highly useful to perform:*

L334-336: "Further studies of purified cytochrome *bcc-aa₃* supercomplex and cytochrome *bd* oxidase would be useful to confirm the differential CO susceptibility of these enzymes and mechanisms of CO inhibition."

The final lines of Results (262-265) are highly speculative and not useful.

We agree with the reviewer and have removed these lines from the revised manuscript.

References:

1. Snapper SB, Melton RE, Mustafa S, Kieser T, Jr WRJ. 1990. Isolation and characterization of efficient plasmid transformation mutants of *Mycobacterium smegmatis*. *Molecular Microbiology* 4:1911-1919.
2. Berney M, Greening C, Conrad R, Jacobs WR, Cook GM. 2014. An obligately aerobic soil bacterium activates fermentative hydrogen production to survive reductive stress during hypoxia. *Proceedings of the National Academy of Sciences* 111:11479-11484.
3. Cordero PR, Grinter R, Hards K, Cryle MJ, Warr CG, Cook GM, Greening C. 2019. Two uptake hydrogenases differentially interact with the aerobic respiratory chain during mycobacterial growth and persistence. *Journal of Biological Chemistry* 294:18980-18991.
4. Tyanova S, Temu T, Sinitcyn P, Carlson A, Hein MY, Geiger T, Mann M, Cox J. 2016. The Perseus computational platform for comprehensive analysis of (prote)omics data. *Nature Methods* 13:731-740.
5. Shiloh MU, Manzanillo P, Cox JS. 2008. *Mycobacterium tuberculosis* senses host-derived carbon monoxide during macrophage infection. *Cell host & microbe* 3:323-330.
6. Kumar A, Deshane JS, Crossman DK, Bolisetty S, Yan B-S, Kramnik I, Agarwal A, Steyn AJ. 2008. Heme oxygenase-1-derived carbon monoxide induces the *Mycobacterium tuberculosis* dormancy regulon. *Journal of Biological Chemistry* 283:18032-18039.
7. Wareham LK, Begg R, Jesse HE, Van Beilen JW, Ali S, Svistunenko D, McLean S, Hellingwerf KJ, Sanguinetti G, Poole RK. 2016. Carbon monoxide gas is not inert, but global, in its consequences for bacterial gene expression, iron acquisition, and antibiotic resistance. *Antioxidants & redox signaling* 24:1013-1028.
8. Jesse HE, Nye TL, McLean S, Green J, Mann BE, Poole RK. 2013. Cytochrome bd-I in *Escherichia coli* is less sensitive than cytochromes bd-II or bo³ to inhibition by the carbon monoxide-releasing molecule, CORM-3: N-acetylcysteine reduces CO-RM uptake and inhibition of respiration. *Biochimica et Biophysica Acta (BBA)-Proteins and Proteomics* 1834:1693-1703.
9. Southam HM, Smith TW, Lyon RL, Liao C, Trevitt CR, Middlemiss LA, Cox FL, Chapman JA, El-Khamisy SF, Hippler M. 2018. A thiol-reactive Ru (II) ion, not CO release, underlies the potent antimicrobial and cytotoxic properties of CO-releasing molecule-3. *Redox biology* 18:114-123.
10. Forte E, Borisov VB, Siletsky SA, Petrosino M, Giuffrè A. 2019. In the respiratory chain of *Escherichia coli* cytochromes bd-I and bd-II are more sensitive to carbon monoxide inhibition than cytochrome bo₃. *Biochimica et Biophysica Acta (BBA)-Bioenergetics* 1860:148088.
11. Cordero PR, Bayly K, Leung PM, Huang C, Islam ZF, Schittenhelm RB, King GM, Greening C. 2019. Atmospheric carbon monoxide oxidation is a widespread mechanism supporting microbial survival. *The ISME journal* 13:2868-2881.

February 4, 2021

Prof. Chris Greening
Monash University
Department of Microbiology, Biomedicine Discovery Institute
Innovation Walk
Melbourne, VIC 3800
Australia

Re: mSystems01292-20 (Mycobacteria tolerate carbon monoxide by remodeling their respiratory chain)

Dear Prof. Chris Greening:

The reviewers' assessment of the resubmitted manuscript were very favorable. Although some additional experiments are recommended by reviewer #4, I believe the comments may be addressed in the text of the resubmission. I look forward to receiving your revised manuscript soon.

Below you will find the comments of the reviewers.

To submit your modified manuscript, log onto the eJP submission site at <https://msystems.msubmit.net/cgi-bin/main.plex>. If you cannot remember your password, click the "Can't remember your password?" link and follow the instructions on the screen. Go to Author Tasks and click the appropriate manuscript title to begin the resubmission process. The information that you entered when you first submitted the paper will be displayed. Please update the information as necessary. Provide (1) point-by-point responses to the issues raised by the reviewers as file type "Response to Reviewers," not in your cover letter, and (2) a PDF file that indicates the changes from the original submission (by highlighting or underlining the changes) as file type "Marked Up Manuscript - For Review Only."

Due to the SARS-CoV-2 pandemic, our typical 60 day deadline for revisions will not be applied. I hope that you will be able to submit a revised manuscript soon, but want to reassure you that the journal will be flexible in terms of timing, particularly if experimental revisions are needed. When you are ready to resubmit, please know that our staff and Editors are working remotely and handling submissions without delay. If you do not wish to modify the manuscript and prefer to submit it to another journal, please notify me of your decision immediately so that the manuscript may be formally withdrawn from consideration by mSystems.

Sincerely,

Cynthia Collins

Editor, mSystems

Journals Department
Reviewer comments:

Reviewer #2 (Comments for the Author):

The ms by Bayly is a revision of a previously submitted manuscript. In the revised manuscript, new experimental data is provided in response to the previous reviewer's critiques. This is a significantly revised manuscript.

In general, most of the concerns have been satisfactorily addressed. However, I have a few minor concerns that must be clarified.

Minor concerns:

Line 70; the authors state that the role of CO in mycobacterial physiology remains controversial. Why is this the case, and what is the basis of the statement? No references are provided.

Line 104; I am not sure that respiratory "toxin" is the appropriate term. Minor concern.

Line 108; reference 29; I suggest the authors provide the primary references.

Lines 129-131; analogous recent studies on pharmacological inhibition of cytochrome BC (Q203) and ATP synthase (bedaquiline) will argue that inhibition of BC reroutes electrons to BD. This seems to be true for CO too. See comment on lines 353-354 below.

Lines 157-158; it will be helpful if the authors provide some context in the statement; for example, what bacteria do these references refer to? Also, I suggest the authors update these references as studies on H₂S and Mtb respiration were recently published.

Lines 194-195; the manuscript naturally progresses from figure 1, 2 up to 3. However, then the authors refer back to figure 1. This is odd; perhaps the authors could find a better solution?

Lines 353-354; I am not convinced that the evidence in this manuscript support this statement. Rather, as stated above, a better interpretation (with less speculation) is that CO mediated

inhibition of (susceptible) cytochrome BC reroutes electrons to (resistant) cytochrome BD, leading to increased respiration in an attempt to restore membrane potential/bioenergetic homeostasis.

Reviewer #4 (Comments for the Author):

Bayly and colleagues present evidence for an underlying mechanism of resistance to CO in mycobacteria. Although the involvement of the cytochrome bd oxidase was shown previously in *E. coli*, this work conclusively shows that it also plays a role in CO resistance in the saprophytic *Mycobacterium smegmatis*. Although the concept is not novel this knowledge might be valuable for the mycobacteria research community, especially for a better understanding of the physiological strategies that pathogenic mycobacteria employ to survive in the host.

Major concerns:

- 1- On the first section of results the authors mention that the prolonged lag phase followed by a similar specific growth rate in the CO treatment condition reflects an adaptation. I tend to agree, but to conclusively show that it is indeed the product of an adaptive mechanism one should have re-diluted the cultures in fresh medium once max OD is reached and check if the lag phase now is shortened or disappears.
- 2- It is not clear why the authors preferred to independently confirm the deletion mutant phenotypes with CRISPRi knockdowns instead of simply complementing the deletion mutants with an intact copy of the corresponding gene. This adds unnecessary complexity to the work and, above all, the phenotypes are not entirely the same. For example, although an explanation is provided for the differences on the lag phase, the knockdown of *cydA* affects the specific growth rate in the presence of CO, while the deletion mutant of *cydAB* does not. Why is that so? I would recommend to complement the deletion mutants and check for phenotype rescue.
- 3- If CO completely inhibits Cyt bc₁-aa₃, how come the specific growth rate is not affected in *DcydAB* in the cultures treated with CO? Shouldn't one expect at least slower growth, like it is observed in the knockdown strain? According to these data, one can only affirm that cyt bd oxidase is only necessary for the adaptation period that occurs during the lag phase and not for growth itself.
- 4- As a follow-up on point 3) is it possible that there are compensatory effects from other terminal oxidoreductases contributing to the absence of an altered specific growth rate in *DcydAB* treated with CO? I would recommend the authors to check if the inhibition of O₂ consumption in *DcydAB* by CO has an impact on PMF.

Minor concerns:

- 1- Introduction would benefit from being more succinct and to the point.
- 2- Is there any explanation on why the DosR regulon is only partially induced upon CO treatment?
- 3- Does the decrease in abundance of iron-acquisition related proteins mean increased levels of intracellular free iron? Could that contribute to some oxidative damage and consequently to the observed increase in lag phase duration?

Response to Editor:

The reviewers' assessment of the resubmitted manuscript were very favorable. Although some additional experiments are recommended by reviewer #4, I believe the comments may be addressed in the text of the resubmission. I look forward to receiving your revised manuscript soon.

We thank the editor for handling the manuscript. As suggested, we have addressed the comments of the reviewers with text changes rather than additional experiments. This reflects that, while the experimental suggestions are good ones, they are not required to substantiate our already robust conclusions that are already supported by the concordant findings of multiple independent experiments. We have additionally made various text clarifications as suggested.

Response to Reviewer comments:

Reviewer #2 (Comments for the Author):

The ms by Bayly is a revision of a previously submitted manuscript. In the revised manuscript, new experimental data is provided in response to the previous reviewer's critiques. This is a significantly revised manuscript. In general, most of the concerns have been satisfactorily addressed. However, I have a few minor concerns that must be clarified.

We thank the reviewer for their constructive comments and helpful suggestions regarding our manuscript.

Minor concerns:

Line 70; the authors state that the role of CO in mycobacterial physiology remains controversial. Why is this the case, and what is the basis of the statement? No references are provided.

We agree with the reviewer that we didn't provide adequate background to support this assertion. The point we were making is that the role of CO in mycobacterial physiology is incompletely understood and there are some inconsistencies in the literature regarding the role of CO in this genus. In the revised manuscript we have modified this sentence on line 68 to:

"Our understanding of the role of CO in the physiology of Mycobacterium also remains incomplete."

This better reflects the state of knowledge in this area. We discuss the knowns and unknowns regarding CO in Mycobacterium in the following paragraphs, and so we don't feel it's necessary to reference this sentence.

Line 104; I am not sure that respiratory "toxin" is the appropriate term. Minor concern.

We agree with the reviewer and have substituted 'toxin' for 'poison' on line 100 of the revised manuscript.

Line 108; reference 29; I suggest the authors provide the primary references.

We agree with the reviewer that references 29 and 31 used here were not the most appropriate. They have been removed from the revised manuscript, and references by Boshoff et. al. (1) and Voskuil et. al. (2) have been added, as these are the papers that generated the dos regulon data summarised by Rustad et. al.(3) (reference 32 in the previous submission) showing that at least 48 proteins are Dos-regulated in M. tuberculosis. We have also included a reference by Leistikow et. al.(4) experimentally showing the importance of DosR during hypoxia-induced dormancy. These are cited on lines 106-107 of the revised manuscript.

Lines 129-131; analogous recent studies on pharmacological inhibition of cytochrome BC (Q203) and ATP synthase (bedaquiline) will argue that inhibition of BC reroutes electrons to BD. This seems to be true for CO too. See comment on lines 353-354 below.

Overall, our results strongly support the conclusion here that "Our results show that M. smegmatis utilizes cytochrome bd oxidase as a primary means of resisting inhibition of its respiratory chain by CO." However, we agree with the reviewer that there are likely two mechanisms underlying the increased respiration through cytochrome bd oxidase in the presence of CO: (i) the increased synthesis of cytochrome bd as shown by the proteomics data and (ii) the rerouting of electrons to cytochrome bd when bcc-aa3 is inhibited as indicated by the respirometry data. We have now noted the second possibility in the revised discussion at what is now line 333:

"It is probable that two mechanisms allow cytochrome bd oxidase to maintain respiratory function in the presence of CO: (i) increased transcription as indicated by the proteomics and (ii) rerouting of electrons to the cytochrome bd oxidase when cytochrome bcc-aa₃ complex is inhibited."

Lines 157-158; it will be helpful if the authors provide some context in the statement; for example, what bacteria do these references refer to? Also, I suggest the authors update these references as studies on H₂S and Mtb respiration were recently published.

We have modified this sentence to include reference to the bacteria used for these studies, now on lines 148-151:

“This is consistent with the established role of cytochrome bd oxidase in resistance to NO, CN, and H₂S in M. tuberculosis and E. coli (23-25, 38), as well as its induction in response to CO and insensitivity to CORM-3 treatment in E. coli (26, 27).”

We have additionally included references to the Forte et al., 2016 and the suggested Saini et al., 2020 papers on H₂S responses in E. coli and Mtb.

Lines 194-195; the manuscript naturally progresses from figure 1, 2 up to 3. However, then the authors refer back to figure 1. This is odd; perhaps the authors could find a better solution?

As discussed in the previous round of revisions, after careful consideration, we have concluded that the current format is the best way to present our data. The current organisation of the figures reflects that the growth curves (figure 1), proteomics (figure 2), and respirometry (figure 3) were each conducted simultaneously for the wild-type and mutant strains to enable direct comparison. Hence, it would not be appropriate to split the figures with one focusing on the wild-type and the other focusing on the mutants.

Lines 353-354; I am not convinced that the evidence in this manuscript support this statement. Rather, as stated above, a better interpretation (with less speculation) is that CO mediated inhibition of (susceptible) cytochrome BC reroutes electrons to (resistant) cytochrome BD, leading to increased respiration in an attempt to restore membrane potential/bioenergetic homeostasis.

We agree with the reviewer's point that, when cytochrome bcc is inhibited, electrons will be rerouted to cytochrome bd and have added a sentence in the discussion regarding this as noted above. However, the point we are making at these lines is that CODH is a high-affinity enzyme that likely functions physiologically at very low CO partial pressures. At these partial pressures, the ratio of CO to O₂ would be sufficiently low that cytochrome bcc would not be significantly inhibited by the CO present. This is a reasonable conclusion based on previously published data and the data we present.

Reviewer #4 (Comments for the Author):

Bayly and colleagues present evidence for an underlying mechanism of resistance to CO in mycobacteria. Although the involvement of the cytochrome bd oxidase was shown previously in E. coli, this work conclusively shows that it also plays a role in CO resistance in the saprophytic Mycobacterium smegmatis. Although the concept is not novel this knowledge might be valuable for the mycobacteria research community, especially for a better understanding of the physiological strategies that pathogenic mycobacteria employ to survive in the host.

We thank the reviewer for their comments regarding our manuscript.

We agree that our findings are important for the mycobacterial community, but they also have some broader use for understanding general bacterial responses to CO. As discussed in the previous round of review, it has not been conclusively demonstrated in any bacterium that cytochrome bd oxidase is inherently resistant to CO. For example, work investigating the resistance of cytochrome bd oxidase to CO in E. coli utilized CO releasing molecules (CORMs), which have non-CO related effects on the cell.

Major concerns:

1- On the first section of results the authors mention that the prolonged lag phase followed by a similar specific growth rate in the CO treatment condition reflects an adaptation. I tend to agree, but to conclusively show that it is indeed the product of an adaptive mechanism one should have re-diluted the cultures in fresh medium once max OD is reached and check if the lag phase now is shortened or disappears.

We have carefully worded the results and discussion to indicate this is an inference rather than a firm conclusion. We agree with the reviewer that re-diluting CO adapted cultures into fresh media is a good way of conclusively demonstrating that adaptation has occurred. However, this experiment is not critical to support the main conclusions of this study, namely that in M. smegmatis cytochrome bd is resistant to CO, it is synthesised in response to the gas, and that it mediates CO resistance.

2- It is not clear why the authors preferred to independently confirm the deletion mutant phenotypes with CRISPRi knockdowns instead of simply complementing the deletion mutants with an intact copy of the corresponding gene. This adds unnecessary complexity to the work and, above all, the phenotypes are not entirely the same. For example, although an explanation is provided for the differences on the lag phase, the knockdown of cydA affects the specific growth rate in the presence of CO, while the deletion mutant of cydAB does not. Why is that so? I would recommend to complement the deletion mutants and check for phenotype rescue.

While we agree that in principle complementation of the terminal oxidase mutants would be ideal, the limitations of mycobacterial genetic systems means such an experiment may prove anything but simple. The use of the CRISPRi knockdown strategy was preferable due to its dependability. The decision to use CRISPRi knockdowns rather than complementation to confirm our terminal oxidase mutant phenotypes was addressed in our response to reviewers in the previous round of review:

“We agree with the reviewer that additional evidence directly supporting the role of the terminal oxidase and dosR gene deletions in the phenotypes we observe will greatly strengthen this work. However, complementation of the terminal oxidase knockouts would likely be very challenging given it would require the addition of two or three genes to replace the deleted structural genes (i.e. cydAB, qcrCAB); in our experience, good expression and synthesis of multiple genes ‘in trans’ is highly challenging to achieve in M. smegmatis, especially for membrane-bound

metalloenzymes. As such, to address this point in the revised manuscript, we have elected to utilize a CRISPRi knockdown strategy to provide an independent line of evidence confirming the phenotype associated with the gene deletions.”

We agree that there are some inconsistencies in the behaviour of the knockout vs knockdown growth, but these reflect that it is necessary to use a higher starting inoculum for knockdown studies (starting OD600 of 0.01 for knockdowns vs 0.0005 for knockout) due to the instability of induction. This precluded observation of adaptation of the strains during lag phase and likely explains the mild differences in growth rate, as the strains were effectively adapting to CO during exponential growth. Nevertheless, both the knockout and knockdown data support our core conclusions that respiratory flexibility enhances CO tolerance of mycobacteria. Overall, the growth curve and respirometry data for the CRISPRi knockdown strains validates the phenotypes for the knockout strains, and provides sufficient evidence to support the main findings of our manuscript.

3- If CO completely inhibits Cyt bc1-aa3, how come the specific growth rate is not affected in Δ cydAB in the cultures treated with CO? Shouldn't one expect at least slower growth, like it is observed in the knockdown strain? According to these data, one can only affirm that cyt bd oxidase is only necessary for the adaptation period that occurs during the lag phase and not for growth itself.

*This is a good question. The cytochrome bd oxidase knockout and knockdown strains were the most strongly affected by CO in terms of lag or growth rate. However, as the reviewer correctly notes, this strain is still able to adapt to CO to achieve a high growth yield. Proteomic analysis of this strain shows that more proteins were upregulated in the presence of CO for the cytochrome bd knockout strain compared to the wild-type and other mutant strains, suggesting that other factors may be deployed to adapt to CO in the absence of cytochrome bd. However, the relatively poorly annotated state of the *M. smegmatis* proteome makes it difficult to deduce how this might be occurring by looking at upregulated proteins. Additionally, adaptation not directly related to cellular protein composition may be occurring, which would not be captured by our proteome analysis. This would be an interesting topic for future investigation.*

4- As a follow-up on point 3) is it possible that there are compensatory effects from other terminal oxidoreductases contributing to the absence of an altered specific growth rate in Δ cydAB treated with CO? I would recommend the authors to check if the inhibition of O₂ consumption in Δ cydAB by CO has an impact on PMF.

*This is a good suggestion, but does not seem to be case. In the proteomic data, we saw no evidence of other terminal reductases being overproduced in the cytochrome bd oxidase mutant. Mycobacteria only possess two terminal oxidases, which can be individually but not simultaneously knocked out, as is supported by Figure 3. It is unclear whether *M. smegmatis* possesses other functional terminal reductases; whereas *M. tuberculosis* is capable of fumarate and nitrate reduction, such activities cannot be readily reproduced in *M. smegmatis*. In our experience, pmf is unlikely to change except in the cases of severe phenotypes due to the capacity of *M.**

smegmatis to metabolically remodel (e.g. Greening et al., Plos One 2014). We have now added the following in the revised manuscript at line 246:

“However, we did not observe a significant increase in the production of potential alternative terminal reductases (e.g. putative nitrate or fumarate reductases).”

Minor concerns:

- 1- Introduction would benefit from being more succinct and to the point.

The introduction had to cover a lot of ground. We have nevertheless made multiple edits to the introduction in the revised manuscript, especially the final paragraph, to ensure it is more succinct.

- 2- Is there any explanation on why the DosR regulon is only partially induced upon CO treatment?

We agree this is an important question, though we don't have a clear answer. We have noted in the revised manuscript the following in line 157:

“It is unclear why the dos regulon in M. smegmatis is only partially induced by CO; this may reflect the interaction between sensor kinases and CO, or cross-talk between DosR and other regulatory mechanisms.”

- 3- Does the decrease in abundance of iron-acquisition related proteins mean increased levels of intracellular free iron? Could that contribute to some oxidative damage and consequently to the observed increase in lag phase duration?

While there is merit to this line of reasoning, only two proteins related to iron acquisition had decreased abundance when cells were grown in the presence of CO. As many other proteins in M. smegmatis are involved in iron acquisition or are likely to be regulated by cellular iron concentrations, we consider it overly speculative to conclude from our proteomics data that levels of intracellular free iron are increased due to the presence of CO. As we state in the manuscript, we are more confident that intracellular iron concentrations were not dramatically decreased due to complex formation with CO, given this would be expected to lead to generally increased production of iron repressed proteins.

April 14, 2021

Prof. Chris Greening
Monash University
Department of Microbiology, Biomedicine Discovery Institute
Innovation Walk
Melbourne, VIC 3800
Australia

Re: mSystems01292-20R1 (Mycobacteria tolerate carbon monoxide by remodeling their respiratory chain)

Dear Chris,

Your manuscript has been accepted, and I am forwarding it to the ASM Journals Department for publication. For your reference, ASM Journals' address is given below. Before it can be scheduled for publication, your manuscript will be checked by the mSystems senior production editor, Ellie Ghatineh, to make sure that all elements meet the technical requirements for publication. She will contact you if anything needs to be revised before copyediting and production can begin. Otherwise, you will be notified when your proofs are ready to be viewed.

- Minimum resolution of 1280 x 720

- .mov or .mp4. video format
- Provide video in the highest quality possible, but do not exceed 1080p
- Provide a still/profile picture that is 640 (w) x 720 (h) max

We recognize that the video files can become quite large, and so to avoid quality loss ASM suggests sending the video file via <https://www.wetransfer.com/>. When you have a final version of the video and the still ready to share, please send it to Ellie Ghatineh at eghatineh@asmusa.org.

Sincerely,

Jack Gilbert
Editor, mSystems

Journals Department
Figure S4: Accept
Table S1: Accept
Figure S2: Accept
Table S2: Accept
Table S3: Accept
Figure S3: Accept
Figure S1: Accept